# Amoxicillin-Clavulanic Acid Empirical Oral Therapy for the Management of Children with Acute Haematogenous Osteomyelitis

**DOI:** 10.3390/antibiotics9080525

**Published:** 2020-08-18

**Authors:** Elena Serrano, Irene Ferri, Luisa Galli, Elena Chiappini

**Affiliations:** Paediatric Infectious Disease Unit, Meyer Children’s University Hospital, Department of Health Sciences, University of Florence, Viale G.Pieraccini 24, 50139 Florence, Italy; elenaserrano0902@gmail.com (E.S.); irene.ferri@stud.unifi.it (I.F.); luisa.galli@unifi.it (L.G.)

**Keywords:** acute haematogenous osteomyelitis, children, amoxicillin-clavulanic acid

## Abstract

According to the Guidelines of the European Society of Pediatric Infectious Diseases (ESPID), in low methicillin-resistant *Staphylococcus aureus* (MRSA) prevalence settings, short intravenous therapy is recommended in uncomplicated cases of acute haematogenous osteomyelitis (AHOM), followed by empirical oral therapy, preferentially with first/second-generation cephalosporin or dicloxacillin or flucloxacillin. However, several practical issues may arise using some of the first-line antibiotics such as poor palatability or adherence problems. Clinical, laboratory and therapeutic data from children with AHOM hospitalized in one Italian Paediatric Hospital between 2010 and 2019 were retrospectively collected and analyzed. The aim of the study was to highlight the extent of the use and the possible role of amoxicillin-clavulanic acid in the oral treatment of children with AHOM. Two hundred and ten children were included. *S.aureus* was identified in 42/58 children (72.4% of identified bacteria); 2/42 *S.aureus* isolates were MRSA (4.8%). No *Kingella kingae* was identified. Amoxicillin-clavulanic acid was the most commonly used oral drug (60.1%; *n* = 107/178) and it was associated with clinical cure in all treated children. Overall, four children developed sequelae. One (0.9%) sequela occurred among the 107 children treated with amoxicillin-clavulanic acid. Our results suggest that amoxicillin-clavulanic acid might be an option for oral antibiotic therapy in children with AHOM.

## 1. Introduction

Acute haematogenous osteomyelitis (AHOM) is the most common musculoskeletal infection in children [1]. In most cases, AHOM is the consequence of hematogenous diffusion of a bacterial pathogen, commonly occurring in children under five years of age and in males [2]. Other risk factors for AHOM include history of recent trauma, recent febrile episodes or upper respiratory tract infections, prematurity, congenital or acquired immunodeficiency, or sickle cell disease [3]. *Staphylococcus aureus* is the most commonly isolated pathogen, accounting for 70–90% of AHOM culture-positive cases [4]. Other microorganisms include Group-A *Streptococcus pyogenes*, *Streptococcus pneumoniae*, and Gram-negative rods [5]. *Kingella kingae* is an emergent pathogen, particularly in children aged <4 years [6]. Community-acquired methicillin-resistant *S. aureus* (MRSA) infections are increasingly reported in the United States, while low rates are observed in most European countries [7,8]. Moreover, an increasing frequency of Panton–Valentine leukocidin (PVL)-producing *S. aureus* isolates is reported [9,10]. Since microbiological diagnosis is obtained in about one-third of children, most cases undergo an empirical antibiotic therapy, based on individual clinical features, age, and local epidemiological data. In low/intermediate MRSA prevalence settings, antistaphylococcal penicillin (i.e., oxacillin or flucloxacillin), a cephalosporin, or clindamycin are the recommended first-line treatment. Historically, AHOM was treated with intravenous (IV) antibiotics for several weeks [1]. However, accumulating data support the use of short IV therapy followed by oral antibiotic therapy in uncomplicated cases [11,12,13,14,15] since this regimen appears to be as effective as longer antibiotic IV courses with no increased risk of complications or sequelae. Short IV therapy (about seven days) is now adopted by several centers in Europe [14] and the US [15]. The European Society of Paediatric Infectious Diseases (ESPID) guidelines for Bone and Joint infections [1] recommend switching from cefazolin to cephalexin or cefuroxime, IV clindamycin to oral clindamycin, and IV ceftriaxone + oxacillin to oral equivalents, such as dicloxacillin or flucloxacillin. At the same time, amoxicillin-clavulanic acid is considered an alternative option. However, “thorough evidence is lacking, and the tolerance is worse” [1,13,14,15]. However, several practical issues may arise. Some antibiotics considered as the first-choice drug by the ESPID guidelines may have low bone penetration (i.e., cephalexin, cefadroxil), poor palatability, adherence issues with four daily administrations (i.e., dicloxacillin, flucloxacillin), or poor tolerability (i.e., risk of a severe rash or *Clostridium difficile* diarrhea using clindamycin) [16,17,18]. Amoxicillin-clavulanic acid therapy would be more feasible in children. However, only a few studies are available in children with AHOM [4,5,19], and bone penetration varies from 10 to 20% [20,21,22]. The aim of our study was to evaluate the management and outcome of AHOM in a third-level Italian university hospital over ten years, with particular consideration to the possible role of amoxicillin-clavulanic acid in the empirical oral therapy in children with AHOM. 

## 2. Results

Overall, 210 children were included in the study (Table 1).

Median age was 5.5 years. The lower limb was the most frequently affected site (120/210; 57.14%), and the most commonly bones involved were femur (40/210; 19.05%) and tibia (35/210; 16.67%). Fever at onset was observed in 54.29% (*n* = 114) cases. Other frequently observed symptoms were (pain 184/210; 87.62%; functional limitation: 157/011; 74.76%; swelling: 115/210; 54.8%; rubor: 102/210; 48.57%; erythema: 71/210; 33.8%). 

One hundred-sixteen out of 210 children (55.24%) developed a complicated AHOM (Table 1). In particular 73/116 children presented with one complication (62.93%), 26 children two complications (22.41%) and 17 children developed at least three complications (14.66%): specifically, 46/210 (21.90%) arthritis, 35/210 (16.8 16.67%) sub-periosteal abscess, 29/210 (13.81%) cellulitis, 23/210 (10.95%) sepsis or septic shock, 26/210 (12.38%) muscular abscess, 15/210 (7.14%) pathological fracture, 10/210 (4.76%) required admission to an intensive care unit, 5/210 (2.38%) children developed deep vein thrombosis and 2/210 (0.95%) septic emboli.

### 2.1. Inflammatory Indices

Sixty-three point zero percent (*n* = 131/205) of children displayed white blood counts (WBC) < 12,000/µL, and C-reactive protein (CRP) < 10 mg/dL in 84.7 82.69% (*n* = 150 172/177 208) of children CRP < 1 mg/dL 28.36% (*n* = 59/208). Erythrocyte sedimentation rate (ESR) was >20 mm/h in 76.9 76.61% (*n* = 113 131/147 171) of cases. 

### 2.2. Microbiological Tests

Overall, 153/210 (72.86%) children underwent at least one microbiological test (Table 2). At least one organism was isolated in 37.9% (58/153) cases. *S. aureus* was the most commonly identified pathogen (72.4% of identified bacteria, 42/58); 5.7% strains (*n* = 2) were Panton–Valentine leukocidin (PVL) producer, 2/42 *S.aureus* isolates were MRSA (4.8%). No *Kingella kingae* strain was identified.

Blood culture was performed in 85/210 (40.5%) children and yielded positive results in 36/85 (42.4%) cases. Culture on pus/biopsy was performed in 45/210 (21.4%) cases; one pathogen was isolated in 18/45 (40.0%) cases. PCR assay on a blood sample was performed in 119/210 (56.7%) cases and was positive in 8/119 (6.7%) children. PCR assay on pus/biopsy was performed in 53/210 (25.2%) children and was positive in 30/53 (56.6%) children. A positive result both in PCR and culture assays were obtained very few patients. In particular, in 28 children, both culture and PCR were positive and consistent (24 *S. aureus* positive), in 25 children, culture was positive but PCR was negative (21 *S aureus* positive), and, in five, culture was negative and PCR was positive (2 *S aureus* positive results)”.

### 2.3. Imaging Studies

Conventional radiography was performed in 185/210 children (88.1%). A radiographic image compatible with AHOM was obtained in 76/185 (41.1%) children. Magnetic resonance imaging was performed in 197/210 cases (93.8%), and images compatible with a diagnosis of AHOM were obtained in 196/197 cases (99.5%). Computed tomography scan was performed in 16/210 children (7.6%), and images compatible with AHOM diagnosis were observed in 14/16 cases (87.5%). Bone scintigraphy was performed in 8/210 children (3.8%), and results compatible with the diagnosis of osteomyelitis were detected in all cases. 

### 2.4. Antibiotic Therapy

All 210 children received IV therapy. Median duration of IV therapy was 18 days (Table 1**)**. The preferred IV regimen was oxacillin plus third generation cephalosporin (*n* = 130/210; 61.9%), followed by therapy including a glycopeptide (*n* = 22/210 cases; 10.5%), or clindamycin (*n* = 19/210 cases; 9.0%), or clindamycin and teicoplanin (*n* = 2/210; 1.0%), or oxacillin in monotherapy (*n* = 11/210; 4.2%). Other used IV regimens were cefazolin (*n* = 4/210; 1.9%), linezolid (*n* = 3/210; 1.4%), ceftazidime (*n* = 6/210; 2.9%), ampicillin/sulbactam (*n* = 7/210; 3.3%), amoxicillin-clavulanic acid (*n* = 2/210; 1.0%), ceftazidime plus rifampicin (*n* = 2/210; 1.0%), other regimens (*n* = 2; 1.0%) 

In 112/210 (53.3%) children first-line IV therapy was switched to a second-line IV therapy for the following reasons: clinical failure (*n* = 17/210; 8.1%), bacterial identification (*n* = 23/210; 11.0%), adverse event (*n* = 8/210; 3.8%), simplification (*n* = 35/210; 16.7%), antibiotic no longer available in hospital during treatment (*n* = 5/210; 2.4%). In 24/210 cases (11.4%), the reason was not specified. One hundred and seventy-eight children were switched to oral antibiotic therapy (84.7%). The median duration of oral therapy was 16 days (Table 1). 

Amoxicillin-clavulanic acid was the most commonly used oral drug (*n* = 107/178, 60.1%). Other antibiotics used in oral therapy were co-trimoxazole plus rifampicin (*n* = 17/178; 9.6%), clindamycin (*n* = 11/178; 6.2%), and flucloxacillin (*n* = 6/178; 3.4%), cephalexin (*n* = 3/178; 1.7%), linezolid (*n* = 12/178; 6.7%), rifampicin + linezolid (*n* = 2/178; 1.1%), cefixime (*n* = 2/178; 1.1%), cefpodoxime (*n* = 2/178; 1.1%), clindamycin + rifampicin (*n* = 2/178; 1.1%), clindamycin + co-trimoxazole (*n* = 2/178; 1.1%), other regimens (*n* = 9/178; 5.1%). Thirty-nine out 210 children (18.6%) underwent surgery. 

### 2.5. Follow-Up Results 

Seventeen out of 210 children (8.1%) were switched to second-line IV therapy for clinical failure, and all of them resolved after the second-line treatment. Treatment failure with the need of rehospitalization was observed in five children (2.4%), and three of these children were treated with amoxicillin-clavulanic acid (3/107; 2.8%). Four children (1.9%) developed sequela including two children with angular deformity and two with limited articular movement. One (0.9%) sequela occurred among the 107 children treated with amoxicillin-clavulanic acid. 

## 3. Discussion

The present study is a continuation of a previous one published in 2017 [23] and confirm other epidemiological data present in literature [24]. As expected, the microbiological diagnosis was reached in about one third of children, and *S.aureus* was the most commonly identified pathogen, being about 70% of the total. Moreover, the prevalence of MRSA infection is confirmed to be low in our setting (only 2/42 *S. aureus* isolates). Prevalence of MRSA strains was in line with those previously reported in Europe. In the United States the incidence of community-acquired MRSA (CA-MRSA) is increased [25], but in Europe CA-MRSA is still uncommon and below 2% [26]. The reason for the low number of positive blood PCR in comparison to the positive blood culture in our study is unclear and more studies are needed at this regard.

The preferred IV antibiotic regimen was oxacillin plus third-generation cephalosporin, adopted in more than 60% of children, while the most commonly prescribed oral antibiotic was amoxicillin-clavulanic acid, which was prescribed in a similar rate (60%). Our data demonstrate the extensive use of amoxicillin-clavulanic acid (>60%). Probably physicians considered its good activity against methicillin-sensitive *S. aureus* (MSSA), pharmacokinetics/pharmacodynamics profile and the low severe event event rate [16,27]. Interestingly, among children treated with amoxicillin-clavulanic acid, only three out of 107 needed rehospitalization, and only one (0.9%) sequela occurred. Similarly, in a recent Danish study, amoxicillin-clavulanic acid was administered to 82 children (42 children with osteomyelitis [OM] 40 children with SA) for three weeks after IV cefuroxime for at least three days [13]. Relapse was reported in two children (2%) with AHOM, and permanent sequelae were observed in two children (2%) [13] (Table 3). In Filleron et al. study, 176 children were included in the retrospective study: 56 with OM, 95 with SA and 25 with both OM and SA. Oral therapy with amoxicillin-clavulanic acid was administered in 82/176 children (46.6%) [14]. Sequelae were observed in two children (1%) [14] (Table 3). Roul-Levy et al. retrospectively compared two groups of children treated for AHOM (*n* = 45) [19]. One group was treated with oral amoxicillin-clavulanic acid (without IV therapy), and the other group had received two to four days of IV therapy (Cefamandolo) followed by oral amoxicillin-clavulanic acid for four weeks. Treatment failure was observed in four children (8.9%) (one in the oral group and three in the IV group) [19] (Table 3). Complexly, these results evidence a large use of amoxicillin-clavulanic acid in children with AHOM, although this regimen is not the first-choice therapeutic option for oral therapy, according to the international guideline recommendations. Apparently, this drug seems to be administered to young children more easily than other molecules recommended as first line therapy. Available data suggest good efficacy. 

Our study has several limitations due to its retrospective nature. Moreover, 31% of children did not underwent any microbiological tests. Another limitation is that the prevalence of MRSA varies from region to region, and therefore the observed low prevalence may not be the same throughout the nation. Another limitation is that a PCR using specific primers for *K. kingae* with a higher sensitivity than the broad-range 16S rDNA real-time PCR used in this study was available only starting from 2019 (performed only in 10 children). In addition, *K. kingae* is a fastidious to grow bacterium. This may explain the fact that no *K. kingae* was identified in our dataset.

In conclusion, in our dataset amoxicillin-clavulanic acid was used in two-thirds of children and associated with clinical cure and no failure in all of them. This finding is confirmed in larger studies [13,14,19], and suggests that amoxicillin-clavulanic acid might by recommended for oral antibiotic therapy in children with AHOM.

## 4. Methods

### 4.1. Definitions 

AHOM was defined as any bone infection with a period between symptoms onset and diagnosis < two weeks [28,29]. Osteomyelitis was diagnosed in the presence of clinical (fever, swelling, warmth, pain, movements limitations) and radiological signs compatible with AHOM with or without bacterial isolation [26]. 

AHOM was defined complicated in the presence of sepsis, septic shock, arthritis, cellulitis, sub-periosteal or muscle abscess, deep vein thrombosis [26] pathological fracture, septic emboli or hospitalization in the intensive care unit [30]. 

### 4.2. Study Design and Population 

A retrospective study was conducted in a single tertiary center, evaluating data of all children aged between one month and 18 years referred to the Meyer Paediatric University Hospital, Florence, Italy, from 1 January 2010, to 31 December 2019, with discharge code with the diagnosis of osteomyelitis (ICD code 10 M86.00-86.99), following The World Health Organization International Classification of Disease (WHO ICD-10). Inclusion criteria were: diagnosis of AHOM, as defined above; age between one month and 18 years. Exclusion criteria were: age ≤ 30 days, congenital or acquired immunodeficiency, underlying bone disease, carrier of prosthetic material, open fracture or surgery at the infection’s site or in an adjacent area, isolated septic arthritis (SA) (with no evidence of osteomyelitis in the adjacent area), hospital-acquired infections. The study is a continuation of a previous study published in 2017 [23]. Clinical, laboratory and therapeutic data of children with AHOM were collected and analyzed, as previously described [23]. In particular, clinical specimens for cultural analysis were collected into transport vials and inoculated onto an array of culture media suitable for detection of bacteria, and fungi (Columbia blood agar, chocolate agar, Schaedler CNA agar, Shaedler KKV agar, and sabouraud dextrose agar) after an enrichment step at 35 °C in thioglycollate broth and nutrient broth. Cultures were carried out at 35 °C for 48 h under aerobic (Columbia blood agar and sabouraud dextrose agar), 5% CO_2_ enriched (chocolate agar) or anaerobic (Schaedler CNA agar and Shaedler KKV agar) conditions. Identification of microbial isolates and antimicrobial susceptibility testing were carried out by the Vitek2 automated system (bioMérieux, Craponne, France).

Broad-range 16S rDNA real-time PCR followed by PCR product sequencing was performed as previously prescribed [23]. Bacterial genomic DNA was extracted from 200 µL of biological samples using the QIAmp DNA Easy Blood & Tissue kit (Qiagen, Venlo, The Netherlands), according to the manufacturer’s instructions. RT-PCR for several bacteria (*Staphylococcus aureus*, *Streptococcus pyogenes*, *Pseudomonas aeruginosa*, Kingella kingae) were performed using specific primers and probes, as previously described [23]. All reactions were performed in triplicates. A negative control (no template) and a positive control for each pathogen were included in every run. DNA was amplified in an ABI 7500 sequence detection system (Applied Biosystems, Foster City, CA, USA, brand of ThermoFisher, Waltham, MA, USA) using the following cycling parameters: 95 °C for 10 min followed by 45 cycles of a two-stage temperature profile of 95 °C for 15 s and 60 °C for 1 min. 

## Figures and Tables

**Table 1 antibiotics-09-00525-t001:** Characteristics of the 210 study children.

Sex (*n*, %)	Female	83/210 (39.52%)
	Male	127/210 (60.48%)
Median age (years; IQR)		5.5 (2–11)
Biopsy executed (*n*,%)		33/210 (15.71%)
Positive blood PCR (*n*,%)		8/119 (6.72%)
Positive pus/biopsy PCR (*n*,%)		30/53 (56.60%)
Positive blood culture (*n*,%)		36/85 (42.35%)
Positive pus/biopsy culture (*n*,%)		23/45 (51.11%)
*Staphylococcus aureus* infection (*n*,%)		42/210 (20.00%)
IV therapy; days (median, IQR)		18 (10–23)
IV therapy < 7 days (*n*;%)		28/210 (13.33%)
IV therapy ≥ 7 days (*n*;%)		182/210 (86.67%)
Oral therapy; days (median, IQR)		16 (13–32)
Total therapy; days (median, IQR)		41 (28–58)
IV drugs	Single therapy	9/210 (9.05%)
	Combination therapy	191/210 (90.95%)
Oral therapy	Single therapy	130/210 (61.90%)
	Combination therapy	45/210 (21.43%)
No oral switch		35/210 (16.67%)
Complicated AHOM	Arthritis	46/210 (21.90%)
	Sub-periosteal abscess	35/210 (16.67%)
	Cellulitis	29/210 (13.81%)
	Sepsis or septic shock	23/210 (10.95%)
	Muscle abscess	26/210 (12.38%)
	Pathologic fracture	15/210 (7.14%)
	Intensive care unit	10/210 (4.76%)
	Deep venous thrombosis	5/210 (2.38%)
	Septic emboli	2/210 (1.09 0.95%)

Note. IV: intravenous; IQR: interquartile range.

**Table 2 antibiotics-09-00525-t002:** Isolates in at least one microbiological test.

Organism	Number of Isolated Organisms	Positive Blood Culture (*n*; %)	Positive Pus/Biopsy Culture (*n*; %)	Positive Blood PCR (*n*; %)	Positive Pus/Biopsy PCR (*n*; %)
*Staphylococcus aureus*	42	21 (72.41%)	14 (77.78%)	1 (33.33%)	20 (80.00%)
*Streptococcus pyogenes*	7	2 (5.56%)	3 (13.04%)	1 (12.5%)	4 (13.33%)
*Proteus mirabilis*	2	0 (0.00%)	2 (8.70%)	0 (0.00%)	0 (0.00%)
*Pseudomonas aeruginosa*	2	0 (0.00%)	1 2 (5.56 8.70%)	0 (0.00 %)	1 (3.33%)
*Streptococcus agalactiae*	2	1 (2.78%)	0 (0.00%)	1 (33.33 12.5%)	0 (0.00%)
*Fusobacterium necrophorum*	1	0 (0.00%)	0 (0.00%)	0 (0.00%)	1 (3.33%)
*Streptococcus pneumoniae*	1	0 (0.00%)	0 (0.00%)	0 (0.00%)	1 (3.33%)
*Staphylococcus epidermidis*	2	2 (5.56%)	0 (0.00%)	0 (0.00%)	0 (0.00%)
*Escherichia coli*	1	0 (0.00%)	1 (4.35%)	0 (0.00%)	0 (0.00%)
*Staphylococcus hominis*	1	1 (2.78%)	0 (0.00%)	0 (0.00%)	0 (0.00%)
*Veilonella parvula*	1	1 (2.78%)	0 (0.00%)	0 (0.00%)	0 (0.00%)
*Streptococcus mitis*	1	1 (3.45%)	0 (0.00%)	0 (0.00%)	0 (0.00%)
*Staphylococcus simulans*	1	0 (0.00%)	1 (4.35%)	0 (0.00%)	0 (0.00%)
*Corynebacterium amycolatum*	1	0 (0.00%)	1 (4.35%)	0 (0.00%)	0 (0.00%)

**Table 3 antibiotics-09-00525-t003:** Recent studies on the management of children with osteomyelitis and use of amoxicillin-clavulanic acid (amoxiclav).

Study	Total	OM	Median Age (Years)	IV Therapy	Duration IV Therapy (Days)	Oral Amoxiclav (*n*, %)	Dosage (mg/kg Per Day)	Duration oral Therapy (Days)	Sequelae (*n*, %)	Relapse (*n*, %)	*S. aureus* (*n*, %)
Our data	210	210	5.5	Oxacillin + 3rd generation cephalosporin (*n* = 130, 61.9%), other (*n* = 80, 38.1%)	18	107/178 (60.1%)	80	16	4 (1.9%) (1 in oral amoxiclav group)	5 (2.4%) (3 in oral amoxiclav group)	42 (72.4%)
Nielsen et al. 2019 [13]	82	42	4.9	Cefuroxime	6 (OM), 4 (SA)	82/82 (100%)	-	19 (OM) 11 (SA)	2 (2.4%)	2 (2.4%)	23 (51%)
Filleron et al. 2019 [14]	176	56	1.8	Cloxacillin or amoxiclav +/− gentamicin	4	82/176 (46.6%)	-	11	2 (1.1%)	-	30 (41%)
Roul-Levy et al. 2016 [19]	45	45	1.6	Cefamandole (*n* = 26)	2–4 (IV group)	45/45 (100%) (19 exclusive oral therapy; 26 2–4 day IV therapy before oral switch)	80		4 (8.9%) (3 in IV group; 1 in oral group)	-	-

IV: intravenous; OM: osteomyelitis; SA: septic arthritis; amoxiclav: amoxicillin-clavulanic acid.

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
