# Peer review of "Amoxicillin-Clavulanic Acid Empirical Oral Therapy for the Management of Children with Acute Haematogenous Osteomyelitis"

_antibiotics, 2020, doi:10.3390/antibiotics9080525_

Round 1

Reviewer 1 Report

The reviewer would like to thank the authors for their efforts in collecting the data and writing the manuscript. Some points had raised during the reviewing of the manuscript which need to be addressed.

The authors had collected the data about an important topic which is the reatment of acute haematogenous osteomyelitis in children using antibiotics. Osteomyelitis is one of the serious diseases that might lead to severe side effects but not really dealt with recently in the literature.

INTRODUCTION

The introduction needs to be rewritten giving more details about AHOM and the different line of treatments in particularly with antiobiotics.

The aim of the study should be mentioned in the last paragraph in the introduction not in the methods section.

The authors had mentioned that this study is part of a previous study. Then why they have used the data from the other study again. It would have been beneficial if they had included the data from January 2016. This study would be a duplicate of the other one. The authors either focus on the treatment of AHOM for the 9 years including the micro-organisms involved and the types of antibiotics administrated but it would be much more worthy if they collect all the demographic data, treatment and antibiotic treatment from January 2016.

Reviewer 2 Report

The manuscript of Serrano et al describes data on amoxicillin-clavulanic acid empirical oral therapy for the management of children with acute haematogenous osteomyelitis.

This manuscript targets on the outcome of amoxicillin clavulanic therapy for the treatment of haematogenous osteomylitis. Although this is a relevant topic several issues need  substantial clarification.

Comments:

  1. The manuscript requires correction for English.
  2. Abstract line 23: The authors state “Our results seem to indicate a possible role of…”. This is very vague and formulated like it is, this sentence does not provide any information. What do the authors want to tell us? Please make a proper statement; to what role are the authors referring? Please specify.
  3. Abstract line 21, and line 93. The authors make a remark that “No Kingella kingae was isolated”. This remark needs a further explanation. Kingella kingae is a known cause of osteomyelitis in children and a fastidious to grow bacterium, that possibly could explain the lack of Kingella kingae culture in this study. Furthermore, Kingella kingae is often identified from biopsies by PCR testing. Could the PCR that was applied in 43 cases of this study (Table 1) detect Kingella kingae?
  4. In the Methods section (and the whole manuscript) any description of the PCR test applied in this study is lacking. Without this information the PCR results as presented in Table 1 cannot be interpreted. This has to be solved by the authors. What PCR test was used, what does it detect and what is the sensitivity and specificity of this test.
  5. Table 1. Were the PCR results congruent with the culture results? Precise information on the PCR test results is required for making any interpretation.
  6. Line 91: It is stated that at least one organism was isolated in 48 cases. However, in Table 1 29 positive blood cultures and 18 positive pus/biopsy culture; that makes 47 in total. Please clarify the discrepancy.
  7. Table 2: Provide the full name for the bacterial species listed.
  8. Table 2: Do the authors have an explanation for the low number of positive blood PCR in comparison to the positive blood culture? Please comment.
  9. Antibiotic therapy. What antibiotic therapy was applied for the single MRSA case? What was the course of antibiotic therapy?
  10. Line 118: Please correct “233/184” to “23/184”
  11. Lines 124-129, Section 3.5. Follow-up results: Please provide more information on these cases; are there any explanations for the clinical failures? For example, was one of these cases the MRSA case?
  12. Line 132: What results were confirmed that are described in reference 28
  13. Discussion, lines 142-151. Of relevance to mention is that in the studies of Filleron et al. 2019 and Nielsen et al 2019, the OM cases were part of their study, 56/176 cases and 42/82 cases, respectively.
  14. Line 156. The conclusion is meaningless. Please provide a conclusion based on the results provided in this study.

Reviewer 3 Report

The present study aimed to retrospectively evaluate the management of acute haematogenous osteomyelitis in a 9-year period in an Italian university hospital, considering especially the role of amoxicillin-clavulanic acid.

A few changes are needed, as follows:

Abstract: Please rephrase the first two sentences! My suggestion “According to the Guidelines of the European Society of Pediatric Infectious Diseases (ESPID), short intravenous therapy is recommended in uncomplicated cases of Acute Haematogenous Osteomyelitis (AHOM), followed by empirical oral therapy….”

Introduction lacks information about acute haematogenous osteomyelitis, its prevalence and causes in children.

Methods: The paragraphs with definitions and complicated AHOM belong to Introduction.

Discussion: Please start with your findings. Please emphasize what is new in your study and which are the implications of your study for clinical practice!

Round 2

Reviewer 1 Report

The reviewer would like to thank the authors for their efforts. However, still the whole introduction is not written in a good way. Alot of paragraphs have the same reference number which means that the authors had just copied and pasted the paragraphs from one source. Another paragraph has 12 references which is too much for the paragraph. Three or four citations would be more than enough.

Author Response

The reviewer would like to thank the authors for their efforts. However, still the whole introduction is not written in a good way. Alot of paragraphs have the same reference number which means that the authors had just copied and pasted the paragraphs from one source. Another paragraph has 12 references which is too much for the paragraph. Three or four citations would be more than enough.

REPLY. We modified the introduction as required and references have been changed.

Reviewer 2 Report

Comments:

  1. Correct the following sentence "Another limitation is that PCR using specific primer for K. kingae was available only starting from 2019. This may explain the fact that no K. kingae was isolated in our dataset." to "Another limitation is that a PCR using specific primers for K. kingae with a higher sensitivity than the broad-range 16S rDNA real-time PCR used in this study was available only starting from 2019. In addition, K. kingae is a fastidious to grow bacterium. This may explain the fact that no K. kingae was identified in our dataset." 
  2. Specify the PCR used at least once in each section of the manuscript as "broad-range 16S rDNA real-time PCR followed by PCR product sequencing". Readers should not always need to look-up references for getting essential information. With specifying the PCR,  readers will directly understand also the meaning of discussing the K. kingae specific PCR.
  3. The authors replied to my comment number 5: " A positive result both in PCR and culture assays were obtained very few patients. In particular in 28 children both culture and PCR were positive and consistent ( 24 S aureus positive), in 25 children culture was positive but PCR was negative (21 S aureus positive) and in 5 culture was negative and PCR was positive ( 2 S aureus positive results)" Please incorporate this important information in the manuscript.
  4. Please also incorporate your reply to my comment 8 in the discussion section.

Author Response

  1. Correct the following sentence "Another limitation is that PCR using specific primer for K. kingae was available only starting from 2019. This may explain the fact that no K. kingae was isolated in our dataset." to "Another limitation is that a PCR using specific primers for K. kingae with a higher sensitivity than the broad-range 16S rDNA real-time PCR used in this study was available only starting from 2019. In addition, K. kingae is a fastidious to grow bacterium. This may explain the fact that no K. kingae was identified in our dataset." 

REPLY. The sentence has been modified as requested

  2. Specify the PCR used at least once in each section of the manuscript as "broad-range 16S rDNA real-time PCR followed by PCR product sequencing". Readers should not always need to look-up references for getting essential information. With specifying the PCR,  readers will directly understand also the meaning of discussing the K. kingae specific PCR.

REPLY. The sentence has been modified as requested. "broad-range 16S rDNA real-time PCR followed by PCR product sequencing" has been added in the method section.

3. The authors replied to my comment number 5: " A positive result both in PCR and culture assays were obtained very few patients. In particular in 28 children both culture and PCR were positive and consistent ( 24 S aureus positive), in 25 children culture was positive but PCR was negative (21 S aureus positive) and in 5 culture was negative and PCR was positive ( 2 S aureus positive results)" Please incorporate this important information in the manuscript.

REPLY We added this sentence to our manuscript, as requested

4. Please also incorporate your reply to my comment 8 in the discussion section.

REPLY an explanation for the low number of positive blood PCR in comparison to the positive blood culture is not clear. We feel that any attempt would be merely a speculation, but we added a comment on this matter in the discussion section and underlined  that “the reason  for the low number of positive blood PCR in comparison to the positive blood culture is unclear and more studies are needed at this regard”